# Dual-Mode Antibacterial Orthodontic Composite: Contact-Killing QACs and Sustained CHX Release via Large-Pore Mesoporous Silica Nanoparticles

**DOI:** 10.3390/ijms26136172

**Published:** 2025-06-26

**Authors:** Xiaotian Teng, Yingguang Cao, Jing Mao, Xiaojuan Luo

**Affiliations:** 1Department of Stomatology, Tongji Hospital, Tongji Medical College, Huazhong University of Science and Technology, Wuhan 430030, China; d2022808807@hust.edu.cn (X.T.);; 2School of Stomatology, Tongji Medical College, Huazhong University of Science and Technology, Wuhan 430030, China

**Keywords:** large-pore mesoporous silica, quaternary ammonium salt, chlorhexidine, dual-mode antibacterial, orthodontic composite

## Abstract

This study develops a dual-mode antibacterial orthodontic adhesive by integrating quaternary ammonium salt-modified large-pore mesoporous silica nanoparticles (QLMSN@CHX). The material integrates two antibacterial mechanisms: (1) contact killing via covalently anchored quaternary ammonium salts (QACs) and (2) sustained release of chlorhexidine (CHX) from radially aligned macropores. The experimental results demonstrated that QLMSN@CHX (5 wt%) achieved rapid biofilm eradication (near-complete biofilm eradication at 24 h) and prolonged antibacterial activity, while maintaining shear bond strength comparable to commercial adhesives (6.62 ± 0.09 MPa after 30-day aging). The large-pore structure enabled controlled CHX release without burst effects, and covalent grafting ensured negligible QAC leaching over 30 days. The composite demonstrated good biocompatibility with human dental pulp mesenchymal stem cells at clinically relevant concentrations. This dual-mode design provides a clinically viable strategy to combat bacterial contamination in orthodontic treatments, with potential applications in other oral infections. Future studies will focus on validating efficacy in complex in vivo biofilm models.

## 1. Introduction

During fixed orthodontic treatment, plaque accumulation around brackets remains a major cause of enamel demineralization and white spot lesions (WSLs) [1]. Regular cleaning of plaque around the brackets or using antibacterial mouthwashes to prevent *Streptococcus mutans* colonization is essential to preventing WSLs [2]. However, poor brushing compliance in some patients has led to an incidence of WSLs as high as 30% during orthodontic treatment [3]. Therefore, modifying orthodontic composites to incorporate antibacterial properties that can inhibit bacterial colonization is a potential strategy for preventing WSLs.

One approach to modifying composite for antibacterial properties is by incorporating releasable antibacterial agents into the resin matrix, such as silver nanoparticles (AgNPs) [4,5,6], zinc oxide nanoparticles (ZnO) [7,8,9], or chlorhexidine (CHX) [10,11,12], to achieve a sustained antibacterial effect. Releasable antibacterial agents can not only inhibit bacterial growth on the resin surface but also suppress bacteria suspended in the medium far from the resin surface [13]. However, once the antibacterial agents are fully released, their antibacterial performance diminishes, compromising the material’s long-term antibacterial effectiveness [14]. Additionally, excessive release over a short period can increase the material’s cytotoxicity.

Another strategy involves incorporating non-releasable antibacterial agents into the resin to achieve contact-based antibacterial effects. Examples include quaternary ammonium compounds such as methacryloyloxydodecyl pyridinium bromide (MDPB) and dimethylaminododecyl methacrylate (DMADDM) [15,16]. Upon contact with bacteria, these quaternary ammonium compounds interact with bacterial membranes, disrupting protein–lipid interactions, causing leakage of cytoplasmic contents and ultimately leading to bacterial death [17]. These contact-based antibacterial agents exhibit certain antibacterial activity against oral bacteria and biofilms, with minimal impact on the curing performance of the resin [18].

Non-releasable antibacterial agents copolymerize with resin monomers, ensuring that antibacterial efficiency and mechanical properties are maintained over time. However, these agents are less effective in inhibiting bacteria located farther from the resin surface. To combine the benefits of both approaches, researchers have been developing hybrid antibacterial agents that offer dual antibacterial effects, leveraging the strengths of both contact–kill and release–kill mechanisms. For example, Bai et al. developed quaternary ammonium silane-grafted hollow mesoporous silica (QHMS) for metronidazole (MDZ) sustained delivery, resulting in a bimodal contact–kill and release–kill capability [18,19]. Zhang et al. demonstrated that combining MDPB and silver ions resulted in enhanced antibacterial effects compared to MDPB alone [20]. Additionally, nanosilver and 12-methacryloyloxydodecylpyridinium bromide have been used in dental adhesives to inhibit caries [21]. However, in these systems, Ag ions lack effective encapsulation, which can lead to burst release of antibacterial agents.

Mesoporous silica nanoparticles (MSNs) provide an effective solution to this issue by reducing burst release [22]. The mesoporous structure allows the loading of antibacterial agents and offers controlled-release channels, while its modified surface can provide contact-based antibacterial properties [23]. Large-pore silica with radially aligned macropores enable high-capacity drug loading and sustained release, avoiding burst effects [24]. Moreover, when incorporated into resin, MSN forms micromechanical interlocks with the organic matrix, improving the mechanical properties and reducing nanoparticle agglomeration [25].

Chlorhexidine (CHX), a commonly used cationic antibacterial agent, is known for its efficacy in inhibiting *S. mutans*, *Lactobacillus*, and *Actinomyces viscosus* in root surface biofilms [26]. CHX binds to salivary glycoproteins, reducing protein adsorption on tooth surfaces and interfering with plaque formation. Additionally, CHX binds to bacterial extracellular polysaccharides, preventing bacterial adhesion to the acquired pellicle, thus helping to prevent dental caries [27].

Quaternary ammonium compounds (QACs), as widely used antibacterial agents, exert bactericidal effects by altering bacterial membrane potential and permeability via their long alkyl chains, leading to cytoplasmic leakage and eventual cell death. Gong et al. developed an antibacterial polymer matrix with covalently bonded networks by copolymerizing quaternary ammonium methacrylate monomers with resin binders [28]. This covalent immobilization strategy minimizes antibacterial agent loss during long-term use, though its efficacy remains limited against bacteria distant from the resin surface.

To address the limitations of existing materials—such as low drug-loading capacity, rapid agent depletion, and the inability to concurrently achieve contact-killing and release-killing effects—this study designed quaternary ammonium-modified large-pore mesoporous silica nanoparticles (QLMSN@CHX) as functional fillers for orthodontic composites.

The synthesis and validation process initiated with the fabrication of large-pore mesoporous silica nanoparticles (LMSNs), followed by covalent grafting of quaternary ammonium antibacterial groups (QACs) onto LMSN surfaces to produce contact-killing QLMSN. Subsequently, chlorhexidine (CHX) was loaded into the mesopores of QLMSN, culminating in dual-functional QLMSN@CHX nanoparticles that combine contact antibacterial activity (quaternary ammonium groups) with sustained-release antibacterial capacity (CHX). When incorporated into orthodontic resin matrices, QLMSN@CHX enhances antibacterial efficacy through combined mechanisms of surface contact killing and controlled CHX release. Systematic evaluation of bacterial biofilm inhibition confirmed the effectiveness of this dual-mechanism system, with the entire workflow schematically illustrated in Figure 1. By integrating these two antibacterial mechanisms into a single nanoparticle system, QLMSN@CHX addresses the limitations of conventional materials that rely solely on either releasable or contact-based strategies [8,12]. The nanoparticles were incorporated into resin matrices to enhance antibacterial efficacy while maintaining critical mechanical properties, with preliminary assessments confirming stable adhesive performance comparable to commercial products.

## 2. Results

### 2.1. Characterization of Different Nanoparticles (MSN, LMSN, QLMSN, and QLMSN@CHX)

#### 2.1.1. TEM

Transmission electron microscopy images revealed the spherical morphology of MSN. Before pore expansion, MSN is spherical with high density (Figure 2A). After pore expansion, the nanospheres have a diameter of approximately 100–200 nm, with pores radiating from the center to the surface (Figure 2B), consistent with previous research findings [29]. An image of an ultrathin section of epoxy resin-embedded QLMSN prepared with an ultramicrotome (Figure 2C) showed that most of the mesopores were arranged radially from the center to the surface of nanoparticle. After covalent bonding of quaternary ammonium groups and loading chlorhexidine, QLMSN@CHX maintained its original spherical structure (Figure 2D).

#### 2.1.2. FTIR

In the FT-IR spectrum (Figure 3A), two strong peaks at 1087 cm^−1^ and 803 cm^−1^ were observed in LMSN, QLMSN, and QLMSN@CHX, representing the asymmetric and symmetric stretching vibrations of Si-O-Si, respectively [29]. In addition, a broad peak was observed at 3600–3100 cm^−1^, attributed to the adsorption band on the surface of LMSN associated with the stretching vibrational mode of the silanol group [30,31]. The Si-OH stretching vibration was significantly weaker in the spectrum of QLMSN compared to that of LMSN. The two enhanced absorption peaks at 2850 cm^−1^ and 2926 cm^−1^ were assigned to the symmetric and asymmetric stretching vibrations of methylene groups (-CH_2_-), confirming the successful grafting of long alkyl chains. Notably, the characteristic peak of quaternary ammonium salt (C-N^+^ vibration) observed at 1480–1460 cm^−1^ provided evidence for covalent modification with quaternary ammonium compounds (Figure 3A) [19,32]. Furthermore, the absorption peaks at 1650 cm^−1^ could be further interpreted as C-H bending vibration modes in the carbon chains of quaternary ammonium groups [33,34]. Successful loading of chlorhexidine (CHX) was verified by the following characteristic peaks: (1) C=N and C-N stretching vibrations in the 1650–1550 cm^−1^ range, characteristic of the biguanide group; (2) N-H asymmetric stretching vibrations appearing as split double peaks in the 3300–3200 cm^−1^ region, representing typical infrared features of biguanide compounds. These spectral features demonstrated consistency with the fingerprint profile of pure chlorhexidine acetate, confirming the successful loading of chlorhexidine acetate onto the QLMSN carrier (Appendix A) [35].

In the XPS (Figure 3C) full spectrum of QLMSN, peaks for the Si, O, C, and N elements were all detected. The presence of the N peak indicated the successful anchoring of QACs [33].

Thermogravimetric analysis (TGA) (Figure 3B) results showed that LMSN exhibited a mass loss of approximately 8.7%, primarily occurring below 100 °C. This mass loss is attributed to the loss of bound water in LMSN. QLMSN exhibited a mass loss of approximately 24.8%, which is 16.1% higher than that of LMSN. In the first stage, QLMSN exhibited a mass loss at temperatures below 300 °C, primarily attributed to the disruption of physically bound H_2_O molecules through hydrogen bonding and the degradation of methyl groups (-CH_3_) following silane modification [33]. In the second stage, a mass loss was observed between 300 °C and 500 °C, primarily due to the cleavage of covalent bonds between the -Si(OCH)_3_ groups of QACs and the silica nanoparticles [32].

For QLMSN@CHX, the mass loss trend below 500 °C was consistent with that of QLMSN, suggesting that the incorporation of chlorhexidine acetate did not significantly alter the chemical surface structure of QLMSN. The total mass loss of QLMSN@CHX was 32.4%, which is 7.6% higher than that of QLMSN. The mass loss at temperatures exceeding 500 °C may be attributed to the decomposition of chlorhexidine acetate [20] (Figure 3B).

#### 2.1.3. Drug Loading Release and Surface Graft Stability of Modified Macroporous Silica

The stability of the grafted quaternary ammonium groups on mesoporous silica was evaluated using the cumulative release curve of QAC groups separated from QLMSN [36] (Figure 3D). Over 30 days, the cumulative release curve of the quaternary ammonium moieties remained a flat horizontal line. This result indicates that the amount of free QACs did not increase, confirming that the binding of QACs to LMSN was strong and remained stable (Figure 3D). This stable binding state forms the chemical structural basis for the sustained contact antimicrobial properties of QLMSN. The release of CHX increased with the proportion of QLMSN@CHX in the resin. However, the release was slow during the first 10 days and then leveled off to a plateau, indicating that the mesoporous silica voids effectively controlled the release of CHX [11] (Figure 3E).

#### 2.1.4. Antibacterial Property Analysis

The free antibacterial group CHX released from the resin disks diffuses into the bacterial suspension, thereby affecting the viable bacterial concentration in the suspension. In contrast, the contact-based antibacterial group, QACs, predominantly exerts its antibacterial effect through surface contact, impacting the biofilm adhered to the resin surface and thus influencing the viable bacterial concentration in the biofilm eluent. Therefore, performing both plate colony counting and turbidity counting on the two types of bacterial solutions can validate the antibacterial efficacy of CHX and QAC groups.

In the bacterial plate counting experiment, fewer and smaller colonies indicate stronger antibacterial activity. For both the bacterial suspension and the biofilm eluent groups, the blank control group and the LMSN group exhibited the highest colony counts, while the LMSN@CHX and QLMSN groups displayed significantly fewer colonies. Notably, the QLMSN@CHX group showed almost no colonies. This result suggests that LMSN@CHX and QLMSN, as single-component systems, possess certain antibacterial properties, but their efficacy is inferior to that of QLMSN@CHX, which combines both contact-based and release-based antibacterial mechanisms (Figure 4A).

The bacterial turbidity counting method results show that the antimicrobial activity of the single-component LMSN@CHX and QLMSN is weaker than that of QLMSN@CHX (Figure 4B,C), and the difference is statistically significant, consistent with the plate colony counting method (*p* < 0.05). The combined results of the bacterial suspension and bacterial elution liquid preliminarily validate the strong antimicrobial effect of QLMSN@CHX at the macroscopic level.

In the bacteriostatic ring experiment with resin disks, effective inhibition zones appeared within 24 h when QLMSN@CHX was incorporated into the resin at a mass fraction of 2.5%. As the mass fraction of QLMSN@CHX increased, the diameter of the inhibition zone also gradually expanded. A noticeable inhibition zone was observed with 5 wt% QLMSN@CHX (Figure 5A).

The bacterial viability MTT assay results showed that as the mass fraction of QLMSN@CHX incorporated into the resin increased, bacterial activity gradually decreased, indicating a progressive increase in antibacterial effects. Notably, after four days of incubation, the antibacterial efficiency of the resin containing 2.5 wt% QLMSN@CHX was less than 50%, while the antibacterial effects of the 5 wt% and 10 wt% QLMSN@CHX resin disks were comparable, with inhibition rates both exceeding 70% (Figure 5B). This suggests that incorporating 5 wt% QLMSN@CHX into the resin is sufficient to produce an effective antibacterial effect.

The cytotoxicity assay results from co-culturing QLMSN@CHX with human dental pulp mesenchymal stem cells demonstrated that QLMSN@CHX exhibits good biocompatibility. Within 24 h, when the concentration of QLMSN@CHX was below 80 μg/mL, no statistically significant difference in cell viability was observed between the experimental groups and the control group. However, at concentrations of 160 μg/mL and 320 μg/mL, significant differences in cell viability were observed (*p* < 0.05) (Figure 5C).

Based on the results from the bacteriostatic ring experiment, MTT assay, and biocompatibility tests, it can be concluded that incorporating 5 wt% QLMSN@CHX into orthodontic resin not only produces an effective antibacterial effect but also ensures good biocompatibility.

### 2.2. Analysis of Dual-Mode Antimicrobial

Laser confocal microscopy scans of bacterial biofilms show that in the blank control group and LMSN control group (without antimicrobial components), bacterial growth was vigorous from 24 h to 72 h. In contrast, in the LMSN@CHX, QLMSN, and QLMSN@CHX groups, after 24 h of co-culture, all three groups exhibited varying degrees of antimicrobial activity. Among them, the LMSN@CHX group had a relatively higher number of live bacteria at 24 h, while the QLMSN@CHX group contained almost all dead bacteria. After 72 h, the LMSN@CHX and QLMSN groups still had some viable bacteria, whereas in the QLMSN@CHX group, even with extended incubation, the biofilm remained almost entirely composed of dead bacteria (Figure 6A).

Quantitative analysis of the live/dead bacterial fluorescence ratio showed that, after incubation for 24, 48, and 72 h, the LMSN@CHX, QLMSN, and QLMSN@CHX groups all exhibited significant antibacterial effects compared to the LMSN control group and blank group. Among these, the QLMSN@CHX group had the lowest live bacteria ratio at each time point, with statistical significance (*p* < 0.05) (Figure 6B). This trend suggests that while the use of either CHX groups (with free antimicrobial effects) or QAC groups (with contact antimicrobial effects) alone can achieve some degree of antimicrobial efficacy, the combination of both antimicrobial mechanisms results in the most effective antimicrobial activity in the shortest time and significantly inhibits bacterial growth.

### 2.3. Resin Mechanical Properties

#### 2.3.1. Modulus of Elasticity and Bending Performance

Three resin formulations were evaluated for mechanical compatibility: commercial 3M resin (3M Company, Minnesota, MN, USA), unmodified resin (Control), and 5/10 wt% QLMSN@CHX-incorporated resin (Figure 7A–C). Stress–strain profiles of the 5 wt% QLMSN@CHX group closely matched those of the Control and 3M groups (Figure 7A), indicating preserved elastomeric behavior. Bending strength analysis further revealed no significant differences between 5 wt% QLMSN@CHX (78.4 ± 2.5 MPa) and both 3M (76.9 ± 3.5 Mpa, *p* >0.05) and Control (79.4 ± 2.6 Mpa, *p* >0.05). In contrast, the 10 wt% QLMSN@CHX group exhibited significantly higher bending strength (90.5 ± 1.6 Mpa, *p* < 0.05; Figure 7B). These results demonstrate that low nanoparticle loading (5 wt%) preserves the elastomeric behavior of the adhesive, while higher loading (10 wt%) enhances rigidity at the expense of flexibility. The 5 wt% formulation thus achieves an optimal equilibrium between antibacterial functionality (dual-mode antibacterial) and mechanical performance.

#### 2.3.2. Shear Bond Strength

Shear bond strength was assessed for the same three groups (3M, Control, QLMSN@CHX) under immediate and aged conditions (Figure 7C). In immediate shear bond strength testing, the 3M group exhibited the highest bond strength (6.93 ± 0.84 MPa), followed by the Control group (6.71 ± 0.83 MPa) and QLMSN@CHX group (6.26 ± 1.02 MPa), with no statistically significant differences among groups (*p* > 0.05). In aged shear bond strength testing, all groups showed reduced bond strength compared to immediate measurements. However, the QLMSN@CHX group maintained performance comparable to both 3M (6.63 ± 0.09 MPa vs. 6.62 ± 0.10 MPa) and Control (6.60 ± 0.11

MPa), with no significant intergroup differences (*p* > 0.05). These results demonstrate that the incorporation of QLMSN@CHX nanoparticles does not compromise the short- or long-term mechanical integrity of the orthodontic resin.

#### 2.3.3. Water Absorption

The water absorption rates of the adhesive groups are summarized in Figure 7C. The 10 wt% QLMSN@CHX group exhibited the lowest water absorption (12.01 ± 1.046 µg/mm^3^), followed by the 3M commercial adhesive (15.01 ± 1.046 µg/mm^3^), 5 wt% QLMSN@CHX (14.28 ± 1.38 µg/mm^3^), and Control group (15.62 ± 1.10 µg/mm^3^). No statistically significant differences were observed among the 3M, Control, and 5 wt% QLMSN@CHX groups (*p* > 0.05).

#### 2.3.4. Resin Residual Index (ARI)

The Wilcoxon test was used to compare the ARI scores among the three bonding agent groups: 3M commercial composite, unmodified resin (Control), and 5 wt% QLMSN@CHX-incorporated resin. As summarized in Table 1, the incorporation of dual-mode antibacterial nanoparticles (5 wt% QLMSN@CHX) showed no statistically significant effect on bond strength (*p* > 0.05; *n* = 30 per group).

## 3. Discussion

This study demonstrates that using QLMSN@CHX as a nanofiller can impart dual-mode antibacterial properties to orthodontic adhesive resin, providing both contact and releasable antibacterial effects.

Chlorhexidine (CHX) is a commonly used broad-spectrum antimicrobial agent that effectively inhibits the growth of various oral bacteria, including Streptococcus and Lactobacillus species. It is also the most extensively studied nonspecific MMP inhibitor [14]. Many orthodontic patients are advised to use mouthwashes containing CHX to reduce Streptococcus mutans levels, thereby minimizing plaque formation and gingivitis. However, the antimicrobial efficacy of CHX is primarily related to its retention in the oral cavity [37]. Therefore, the incorporation of various slow-release and controlled-release CHX oral antimicrobial delivery systems into orthodontic materials can enhance the retention time of CHX in the oral cavity. For instance, the release of CHX from modified orthodontic elastics can provide a sustained antibacterial effect [38]. Some researchers have developed mesoporous silica nanoparticles carrying two antimicrobial agents, nanosilver and chlorhexidine, to modulate biofilm development towards a non-cariogenic state [39]. Although numerous materials have been developed in previous studies, they still suffer from limitations such as poor drug-loading capacity and the inability to directly contact and eradicate surface-colonized microbial communities. The positively charged nitrogen atom in the quaternary ammonium group interacts electrostatically with the negatively charged bacterial cell wall and membrane, while the lipophilic long-chain alkyl tail inserts into the cell membrane and binds to the protein and lipid layers within it (Appendix A). This disrupts the exchange processes between the bacteria and their external environment, causing leakage of cytoplasmic contents and ultimately leading to bacterial death [40].

Studies on the chain length of quaternary ammonium salts have shown that their antibacterial activity increases with longer chains. However, a threshold exists: when the carbon chain length exceeds 18, the antibacterial activity begins to decline [41]. While the molecular targets of QACs remain debated, their bactericidal action is primarily attributed to membrane disruption via electrostatic and hydrophobic interactions [42]. Sideridou demonstrated that Bis-GMA/TEGDMA containing 2.5–10 wt% silica nanoparticles modified with quaternary ammonium salts exhibits significant antibacterial activity against Streptococcus mutans [43]. To address the issue of depletion in CHX-release-based antimicrobial materials, QLMSN@CHX was developed by covalently linking the -Si-(OCH)_3_ groups of QACs with the -OH groups on the inner and outer surfaces of LMSN through organic chemistry and material science fabrication methods (Appendix A) [44]. Covalently bonding mesoporous silica with QACs is one of the most effective methods for creating stable and reliable contact-active antibacterial materials. This ensures that bioactive compounds do not migrate into the external environment, which is critically important for biomedical applications [45]. Since contact-active antibacterial materials do not release antimicrobial agents, they can maintain their effectiveness even after repeated use and significantly reduce the required dose of active agents to prevent microbial growth.

Compared to traditional small-pore MSNs, LMSNs with expanded pore sizes perform better in drug delivery and diffusion, enabling more efficient drug loading and release [46]. Conventional mesoporous silica (<5 nm pores) cannot simultaneously accommodate CHX loading and QACs grafting without pore blockage, reducing drug capacity [46]. Some studies have reported that the high porosity and large pore volume of LMSNs not only prolong drug release but also effectively reduce bacterial resistance to antibacterial agents [47]. To achieve a sustained-release effect, surface properties are enhanced by introducing functional groups, such as grafting APTES or encapsulating MSN with sustained-release polymers, to modify the surface characteristics and help control the drug release at a specific rate (Appendix A) [44,48]. Therefore, by grafting QACs onto the surface, we can not only functionalize the MSN surface with antimicrobial properties but also modify the surface structure of MSN to facilitate the slow release of antimicrobial drugs [48]. The results also confirmed our prediction that chlorhexidine (CHX) from QLMSN@CHX can be released continuously for 30 days, showing excellent sustained-release properties (Figure 3B). This result is consistent with the findings of Yang et al., which showed that CHX-modified resin effectively inhibited the growth of *S. mutans* biofilms and extended the lifespan of clinical adhesive restorations [27]. Additionally, the advantage of contact-type antibacterial agents lies in the fixed nature of the antibacterial agent, preventing it from being depleted over long-term use [49]. By covalently binding QACs with the surface functional groups of LMSNs, we effectively resolved the reservoir depletion issue associated with CHX-based releasable antibacterial materials [34,35].

When assessing antibacterial performance through bacterial turbidity counting, the LMSN@CHX group exhibited stronger activity than the QLMSN group in the bacterial suspension. This could be attributed to the difficulty of QAC groups in QLMSN detaching from the resin disks, whereas the CHX groups in LMSN@CHX could release and diffuse into the solution, exerting a more potent long-range antibacterial effect on the bacteria in suspension (Figure 4B). In contrast, in the biofilm eluent, the QLMSN group showed stronger antibacterial activity than the LMSN@CHX group, likely due to the QAC groups directly inhibiting bacterial biofilm adhesion on the surface of the resin disks at close range (Figure 4C). Regardless of the bacterial culture method used, it is challenging to distinctly separate the effects of contact and release antibacterial mechanisms. However, the combined action of QACs and CHX significantly enhances antibacterial activity (Figure 4B,C). This result aligns with observations from laser confocal microscopy (Figure 6A).

The antibacterial ring experiment (Figure 5A) showed that as the proportion of QLMSN@CHX increased, the diameter of the antibacterial ring significantly enlarged, demonstrating its excellent antibacterial properties.

The dual-mode antibacterial capability of QLMSN@CHX was thoroughly validated through bacterial biofilm and antimicrobial resin co-culture experiments (Figure 6). After 24 h, the viable bacterial count of LMSN@CHX was higher than that of QLMSN, which may be attributed to the distinct antibacterial mechanisms of the two formulations (Figure 6A). The free antimicrobial group CHX released from LMSN@CHX did not reach sufficient concentration around the biofilm in a short period, failing to form an adequate inhibitory concentration. However, the contact-based antimicrobial group of QACs from QLMSN was able to inhibit biofilm adhesion and suppress most bacterial proliferation through direct contact in the early stages. When used individually, neither LMSN@CHX nor QLMSN could completely prevent bacterial growth at 24, 48, and 72 h (Figure 6A). In contrast, when the two antimicrobial mechanisms were combined, QLMSN@CHX rapidly acted and consistently killed bacteria, with nearly all biofilm cells dead from 24 to 72 h. This demonstrates that the combined effect of contact-based and release-based antibacterial mechanisms can amplify the overall antibacterial efficacy (Figure 6B). This may be because QACs first disrupt the functional membranes on the bacterial surface, allowing the released CHX to more easily pass through the damaged membrane structure and enter the cytoplasm, thereby exerting a strong synergistic antibacterial effect [27,50]. Another possibility is that the chemical properties of QLMSN@CHX surfaces, such as hydrophilicity and surface energy, are altered after quaternary ammonium salt modification, making the surface less favorable for bacterial adhesion. By reducing the initial bacterial adhesion to the surface, biofilm formation can also be effectively inhibited, further enhancing the antibacterial effect [51,52]. However, further research is needed to understand how this dual-mode antibacterial mechanism works synergistically to inhibit bacterial adhesion.

Porous silica has been widely used as a functional filler in resin modification studies because these porous fillers can be infiltrated by liquid resin. The pores enhance micromechanical interlocking between the resin and the filler, thereby improving the mechanical properties of dental resin composites without compromising tensile strength [53,54]. In vitro studies have demonstrated that the incorporation of chlorhexidine into adhesive materials effectively inhibits the activity of various matrix metalloproteinases (MMPs), including MMP-2, MMP-8, and MMP-9. Whether used alone or in combination with functional monomers such as MDP, chlorhexidine delays the degradation of resin–dentin bond strength over time [55,56]. However, our study observed that while higher filler content enhances antibacterial effects, it may also compromise the mechanical properties of the material due to excessive cross-linking between the filler and the resin matrix (Figure 7) [57]. The marginally higher bond strength of the commercial 3M adhesive may stem from optimized industrial formulations. The slight reduction in bond strength post-aging is consistent with typical resin degradation patterns due to hydrolytic and enzymatic challenges in the oral environment. Nevertheless, QLMSN@CHX achieved clinically acceptable performance while providing dual-mode antibacterial functionality—a critical advantage over non-antibacterial commercial products. Therefore, the results indicate that adding 5 wt% QLMSN@CHX is an optimal choice, as it provides sufficient antibacterial efficacy without significantly affecting the mechanical properties of the resin (Appendix A).

As illustrated in Figure 7D, the water absorption rate of the resin gradually decreased with increasing QLMSN@CHX doping ratios, suggesting that both QLMSN and QLMSN@CHX may impart hydrophobic characteristics to the resin matrix. This observation was further supported by TGA (Figure 3C), which revealed that QLMSN@CHX exhibited significantly lower mass loss below 200 °C compared to unmodified LMSN. We hypothesize that the grafting of QAC groups introduces long-chain alkyl moieties (e.g., C_18_) whose hydrophobic properties reduce water adsorption on the nanoparticle surfaces. Previous studies have demonstrated that the hydrophobicity of QACs originates from their long alkyl chains, which can penetrate bacterial lipid membranes through hydrophobic interactions, thereby disrupting membrane integrity and enhancing antibacterial efficacy [58]. However, it should be noted that excessive hydrophobicity may compromise the interfacial wettability between the resin and hydrophilic dental tissues (e.g., dentin), leading to reduced micromechanical interlocking efficiency (as evidenced by the declining bond strength with higher filler loading in Figure 7A). Consequently, the optimal QLMSN@CHX doping ratio must balance antibacterial performance (driven by hydrophobicity) and bond strength (requiring hydrophilicity). In this study, a 5 wt% doping ratio was found to maintain clinically acceptable bond strength while preserving effective antibacterial activity (Appendix A).

As a membrane-targeting compound, QACs are considered a potential candidate drug for combating antibiotic-resistant bacteria [59]. QACs exhibit biofilm-disrupting activity against Gram-positive Staphylococcus aureus and Enterococcus faecalis biofilms [18,60]. With its potent antibacterial properties, QLMSN@CHX could also be utilized in future studies for the treatment of other oral infections, including pulpitis, periapical periodontitis, periodontitis, and peri-implantitis. For example, QLMSN@CHX may serve as an endodontic irrigant in root canal treatments or as a topical medication for periodontal pockets. Additionally, QLMSN@CHX could act as an ideal tissue-engineering substitute to promote bone regeneration in infected wounds. However, despite the progress made in this study, certain limitations remain. For instance, the complex oral ecosystem comprises over 700 bacterial strains, while this study focused solely on Streptococcus mutans, the primary pathogen of dental caries, to better align with orthodontic clinical needs. Therefore, future research should incorporate more complex biofilm models both in vitro and in vivo. Furthermore, as this study was primarily based on in vitro models, future studies should include animal experiments to validate the long-term effects of these materials in vivo [58,61].

Collectively, this study demonstrates that QLMSN@CHX nanoparticles serve as a dual-functional nanofiller for orthodontic adhesives, providing contact killing via surface-bound QACs and sustained CHX release from radially aligned macropores, without compromising mechanical or adhesive properties. However, several limitations should be acknowledged.

The current model focused solely on *S. mutans* monoculture biofilms, whereas the oral cavity harbors polymicrobial communities that may alter antibacterial efficacy. Long-term cytotoxicity and nanoparticle biodistribution in vivo remain to be validated under dynamic oral conditions (e.g., enzymatic activity, pH fluctuations).

Looking forward, the dual-mode antibacterial strategy holds promise beyond orthodontics. For instance, QLMSN@CHX could be adapted as coatings for dental implants to prevent peri-implantitis or be incorporated into periodontal dressings to combat polymicrobial infections. Further exploration of its interactions with host tissues (e.g., promoting mineralization or immunomodulation) may unlock broader therapeutic potential in regenerative dentistry.

## 4. Materials and Methods

**Materials**: All chemicals were purchased from Sigma-Aldrich (Shanghai) Trading Co., Ltd., Shanghai, China) unless otherwise specified: bisphenol glycidyl dimethacrylate (Bis-GMA,), camphorquinone (CQ), triethylene glycol dimethacrylate (TEGDMA), tetraethyl orthosilicate (TEOS), hexadecyltrimethylammonium bromide (CTAB), 1,3,5-trimethylbenzene (TMB), 3-(trimethoxysilyl)-propyldimethyloctadecyl ammonium chloride (QACs), chlorhexidine acetate (CHX), LIVE/DEAD Bacterial Viability Kit(Thermo Fisher Scientific, Waltham, MA, USA), 3M Transbond XT orthodontic Bonding Agent (light cure composite and light cure adhesive primer, 3M Oral Care, St. Paul, MN, USA), methanol, anhydrous ethanol, phosphate-buffered saline (PBS, pH 7.4), Streptococcus mutans UA159(ATCC, Manassas, VA, USA), brain–heart infusion (BHI) broth(BD Biosciences, Franklin Lakes, NJ, USA), and fetal bovine serum (FBS, Gibco, Waltham, MA, USA).

### 4.1. Synthesis of Large-Pore-Size Mesoporous Silica (LMSN)

LMSN was synthesized as previously reported [29]. Briefly, water and methanol were mixed in a 4:6 (*v*/*v*) ratio, and the pH was adjusted to 11. Four grams of CTAB were dissolved in 800 mL of the mixture, followed by rapid addition of 3 mL TEOS. The MSN was redispersed in a 1:1 (*v*/*v*) mixture of deionized water and TMB (total volume: 20 mL) and then hydrothermally treated in a Teflon-lined autoclave at 140 °C for 4 days to achieve mesopore expansion. The surfactant template was subsequently removed by refluxing with a HCl/ethanol mixed solution (1:10 *v*/*v*, total volume: 22 mL) for 20 h. The final product, designated as LMSN, was obtained as a white powder after three ethanol washes and drying at 80 °C for 8 h.

### 4.2. Synthesis of Quaternary Ammonium-Modified Mesoporous Silica (QLMSN)

Two hundred milligrams of LMSN were dispersed in a 20 mL mixture of deionized water and ethanol (1:1, *v*/*v*), followed by the addition of 100 μL of QACs [62]. The mixture was stirred and sealed for 24 h at room temperature. The resulting suspension was washed three times with deionized water and dried to yield QLMSN.

### 4.3. Preparation of Chlorhexidine-Loaded QLMSN (QLMSN@CHX)

Ten milligrams of QLMSN were dispersed in a solution of CHX and ultrasonicated for 10 min. The mixture was stirred at 25 °C for 48 h. Afterward, the suspension was rinsed with deionized water and lyophilized by centrifugation to obtain the QLMSN@CHX powder.

### 4.4. Characterization of LMSN, QLMSN, and QLMSN@CHX

#### 4.4.1. Transmission Electron Microscopy (TEM)

The suspensions were drop-cast onto TEM copper grids, air-dried on filter paper, and subsequently analyzed by transmission electron microscopy (TEM, FEI, Eindhoven, The Netherlands) to characterize their ultrastructure. For QLMSN, after gradient ethanol dehydration and drying, the samples were embedded in epoxy resin, polymerized at 60 °C for 24–48 h, and then ultrathin-sectioned (80–100 nm thickness) using an ultramicrotome. The sections were transferred onto TEM copper grids, air-dried, and imaged by TEM to observe the internal ultrastructure of QLMSN spheres.

#### 4.4.2. Fourier-Transform Infrared Spectroscopy (FTIR)

FTIR spectra of LMSN, QLMSN and QLMSN@CHX powders were analyzed to assess the surface chemical bonds at a spectral resolution of 16 cm^−1^ over a range of 4000–400 cm^−1^ at a scan rate of at least 100 spectra/sec (Thermo Scientific).

#### 4.4.3. Thermogravimetry and Differential Thermal Analysis (TG/DTA)

Thermogravimetric analysis (TGA) of LMSN, QLMSN, and QLMSN@CHX was performed on a Diamond TG/DTA instrument (PerkinElmer, Shelton, CT, USA) under a nitrogen atmosphere, with a heating rate of 10 °C/min from 25 °C to 1000 °C.

#### 4.4.4. X-Ray Photoelectron Spectroscopy (XPS)

The surface chemical composition of LMSN and QLMSN was analyzed using XPS (AXIS-ULTRA DLD-600W, Shimadzu-Kratos, Kyoto, Japan). Full-scan spectra were obtained at 160 eV, and narrow-scan spectra were acquired at 20 eV.

#### 4.4.5. Stability of QLMSN

The quaternary ammonium cation binds to *bromophenol blue* (HBB), forming a blue complex, which decreases absorbance at 616 nm. To assess the stability of the QACs grafted on LMSN, QLMSN was immersed in PBS solution (10 mg/mL) for 30 days. The supernatant was mixed with HBB at different time points (1, 3, 5, 10, 15, 20, and 30 days), and the absorbance at 616 nm was measured. The amount of detached QACs was determined by calculating the absorbance change and plotting the cumulative release curve.

#### 4.4.6. Chlorhexidine Acetate Release from QLMSN@CHX

QLMSN@CHX (10 mg/mL) was immersed in PBS solution for 30 days. At intervals of 1, 3, 5, 10, 15, 20, and 30 days, the CHX concentration in the supernatant was measured at 374 nm using a UV spectrophotometer. The cumulative CHX release was calculated, and a release curve was plotted accordingly.

### 4.5. Cytotoxicity Test

Cytotoxicity to human dental pulp stem cells (HDPSCs) was assessed using the CCK-8 assay kit(purchased via Beyotime Biotechnology, Shanghai, China). HDPSCs were seeded in 96-well plates at a density of 2 × 10⁴ cells per well and cultured for 24 h at 37 °C. Different concentrations of QLMSN@CHX particles served as experimental groups. The medium without nanoparticles was designated as the blank control group, with five replicates per group.

### 4.6. Synthesis of the Composite Resin

The resin matrix was synthesized by combining Bis-GMA (bisphenol glycidyl dimethacrylate), TEGDMA (triethylene glycol dimethacrylate), silica nanoparticles as reinforcing fillers, and the organosilane coupling agent cyclohexane. The method followed protocols from Rodriguez [63], Karabela [43], and Wilson [46]. QLMSN@CHX (2.5 *wt*%, 5 *wt*%, and 10 *wt* %) were added to the resin matrix as fillers. Camphorquinone (CQ, 0.2 *wt*%) and DMAEMA (0.8 *wt* %) were added as photoinitiators. The Bis-GMA-to-TEGDMA ratio was maintained at 50:50 (*wt* %).

### 4.7. Bacteria Inoculation and Biofilm Formation

*S. mutans* UA159 was routinely incubated anaerobically in BHI broth at 37 °C (90% N_2_, 5% CO_2_, 5% H_2_). Biofilm formation was initiated by suspending *S. mutans* to obtain an inoculum containing defined microorganisms (10^8^ colony-forming units [CFUs]/mL) in 2 mL of BHI medium supplemented with 1% sucrose in 24-well plates.

### 4.8. Bacterial Counting Experiment

Resin disks containing different nanoparticles (five groups, *n* = 5 per group) were immersed in 500 μL of *S. mutans* suspension (10⁸ CFU/mL) and incubated for 24 h. After removal of the disks, the remaining planktonic bacterial suspension was collected. Biofilms adherent to the disks were detached by mechanical scraping and ultrasonication and then resuspended in sterile PBS. Both the planktonic suspension and biofilm eluent were serially diluted and plated on BHI agar for CFU quantification. Additionally, bacterial density was assessed spectrophotometrically by measuring optical density at 600 nm (OD_600_), with lower values indicating stronger antibacterial activity.

### 4.9. Agar Diffusion Assay for Antibacterial Activity

The minimum inhibitory concentration (MIC) of QLMSN@CHX was evaluated using an agar diffusion method. *S. mutans* suspensions (10⁶ CFU/mL) were spread on BHI agar plates. Resin disks containing 2.5 wt%, 5 wt%, or 10 wt% QLMSN@CHX (test groups) and blank controls (0 wt% QLMSN@CHX) were placed on the inoculated agar. Plates were incubated anaerobically at 37 °C for 24 h, and zones of inhibition around the disks were measured to assess antibacterial efficacy.

#### 4.9.1. MTT Biofilm Assay

A 1.5 mL suspension of *Streptococcus mutans* was added to a 96-well plate and incubated at 37 °C for 48 h. Fresh BHI medium was replenished every 24 h to facilitate the formation of *S. mutans* biofilms. At various time intervals, the cultured biofilms were transferred to new 24-well plates using sterile forceps. To each well, 1 mL of MTT dye (5 mg/mL) was added, and the plate was incubated at 37 °C for 1 h. After incubation, the biofilms were transferred to a new 24-well plate, and 1 mL of DMSO was added to each well. The plate was placed on a shaker in the dark for 20 min. Subsequently, 300 µL of liquid was extracted from each well, and optical density (OD) was measured at 540 nm to assess the activity of the biofilms and calculate their relative metabolic activity.

#### 4.9.2. Live/Dead Bacterial Detection

Biofilms were stained with 2.5 µM SYTO9 and propidium iodide (PI) for live/dead bacterial detection. Mature *S. mutans* biofilms were incubated for 24, 48, and 72 h, and the distribution of live and dead bacteria was assessed using a confocal laser scanning microscope. Biofilm samples (*n* = 5) were examined along the *Z*-axis. Live bacteria appeared green (SYTO9 staining), and dead bacteria appeared red (PI staining). The layer spacing was 2.92 µm, with a scanning depth of 30 µm, which was optimized based on the biofilm thickness. Five randomly selected fields per sample were analyzed to determine the live/dead cell ratio.

### 4.10. Test of Resin Mechanical Properties

#### 4.10.1. Modulus of Elasticity

Specimens were prepared according to the GB/T 9941-2000 plastic bending performance test method, with dimensions of 20 mm × 2 mm × 2 mm. The specimens were cured using an LED light curing lamp for 60 s. After curing, the specimens were placed in a constant-temperature water bath at 37 °C for 24 h. A three-point bending test was conducted using an electronic universal testing machine (model: CMT6104/ZWICK) at a compression rate of 1 mm/min. Bending strength (MPa), modulus of elasticity (MPa), and strain (%) were recorded.

#### 4.10.2. Test of Shear Bond Strength

This study selected the same bonding agent and brackets but different resin formulations. Premolars extracted for orthodontic treatment from patients aged 18 to 22 years were chosen. Following the standard bonding procedure, the tooth surface was acid-etched, the bonding agent was applied, and the resin was placed on the mesh base of the bracket. The bracket was then positioned on the tooth surface, vertical pressure was applied to extrude excess resin, and each surface was cured for 20 s. All samples were subsequently immersed in deionized water and stored in a thermostat at 37 °C for 24 h for immediate shear bond strength test.

Orthodontic bracket–adhesive–enamel bonding strength was tested by applying a load parallel to the bracket tip at a rate of 0.5 mm/min until the bracket was dislodged. The maximum load (N) and the bracket mesh area (S, approximately 9.84 mm^2^) were recorded. The aged shear bond strength test was assessed after aging the samples in human salivary amylase at 37 °C for 1 month. The bond strength (MPa) was calculated using the formula:Bond strength (MPa) = Maximum load (N)/Base area of the bracket (mm^2^)

#### 4.10.3. Resin Water Absorption

Specimens were prepared according to ISO standards, with a diameter of 15 mm and a thickness of 1 mm. Resin trays were cured at 37 °C and dried for 22 h. Fifty test tubes, each containing 10 mL of deionized water, were prepared, and the specimens were stored for 7 days. After removal, the specimens were blotted dry with filter paper and weighed (W_1_) to the nearest 0.01 mg. The disks were stored at 37 °C and weighed weekly until a constant mass (W_2_) was reached, recorded to the nearest 0.1 mg. Water absorption (WSP) was calculated using the formula:WSP = (W_2_ − W_1_)/V
where V is the volume of the specimen, expressed in µg/mm^3^ [46].

#### 4.10.4. Adhesive Residue Index (ARI)

After the shear bond strength test, the resin residues on the enamel surface were observed using a stereomicroscope. The Adhesive Residue Index (ARI) was recorded. The ARI is an international standard measure of how well the adhesive bonds to the bracket.

### 4.11. Statistical Analysis

Adhesive Residual Index (ARI) scores were determined using the Wilcoxon signed-rank sum test (α = 0.05), with the results reported as median (M), 25th quartile (P25), and 75th percentile (P75). Water absorption, bond strength, and elastic modulus experiments were independently replicated at least three times. One-way analysis of variance (ANOVA) was performed to detect significant effects of variables, with differences considered significant at *p* < 0.05. Statistical analyses were conducted using SPSS software, version 16.0 (SPSS, USA).

## 5. Conclusions

In conclusion, this study establishes a dual-mode antibacterial strategy for orthodontic composites by integrating functionalized large-pore mesoporous silica nanoparticles (QLMSN@CHX). While this proof-of-concept study focused on *S. mutans* as the primary cariogenic pathogen, future work will validate the dual-mode efficacy in clinically relevant mixed-species biofilm models. This dual-mode design provides a clinically viable strategy to combat bacterial contamination in orthodontic treatments, with potential applications in managing other oral infections such as peri-implantitis or periodontal diseases.

## 6. Patents

Certificate number: 6859986; patent number: ZL202210877057.4.

## Figures and Tables

**Figure 1 ijms-26-06172-f001:**
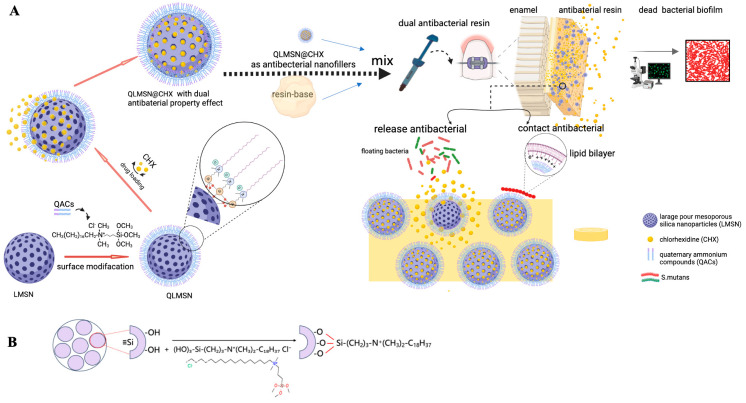
(**A**) Schematic illustration of preparation of nanoparticles with dual-mode antibacterial effect and its application in antibacterial orthodontic resin; (**B**) schematic diagram of the quaternization–functionalization process of large-pore mesoporous silica and molecular structure of the quaternary ammonium salt.

**Figure 2 ijms-26-06172-f002:**
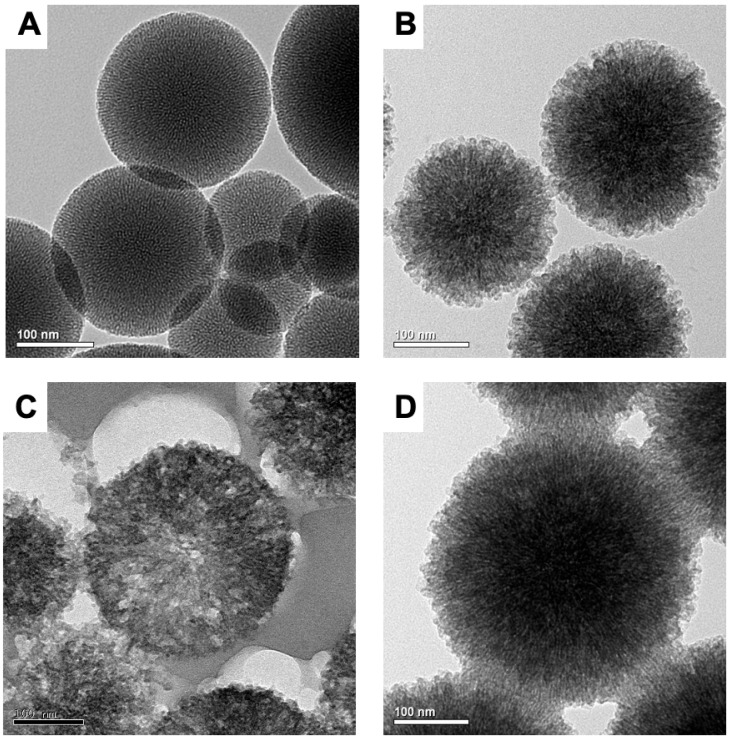
Transmission electron microscopy of different nanoparticles (**A**) MSN; (**B**) LMSN; (**C**) epoxy resin-embedded, sectioned QLMSN (bar: 100 nm); (**D**) QLMSN@CHX (bar: 100 nm).

**Figure 3 ijms-26-06172-f003:**
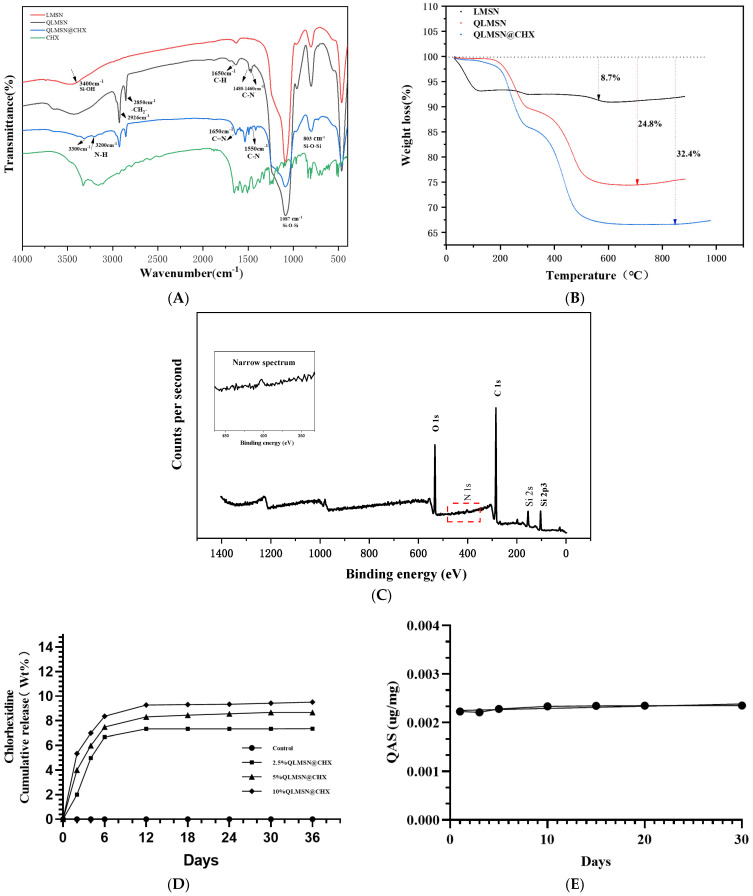
(**A**) FT-IR spectra of LMSN, QLMSN, and QLMSN@CHX. (**B**) TGA comparative scans showing the thermal stability differences among LMSN, QLMSN, and QLMSN@CHX. (**C**) Elemental analysis of QLMSN after QAC grafting. (**D**) Bromophenol Blue Assay illustrating the amount of quaternary ammonium moieties detected in the leachate after storing antibacterial QLMSN nanoparticles in deionized water for 30 days. (**E**) Cumulative QAS release profile of QLMSN@CHX nanoparticles over a 30-day period.

**Figure 4 ijms-26-06172-f004:**
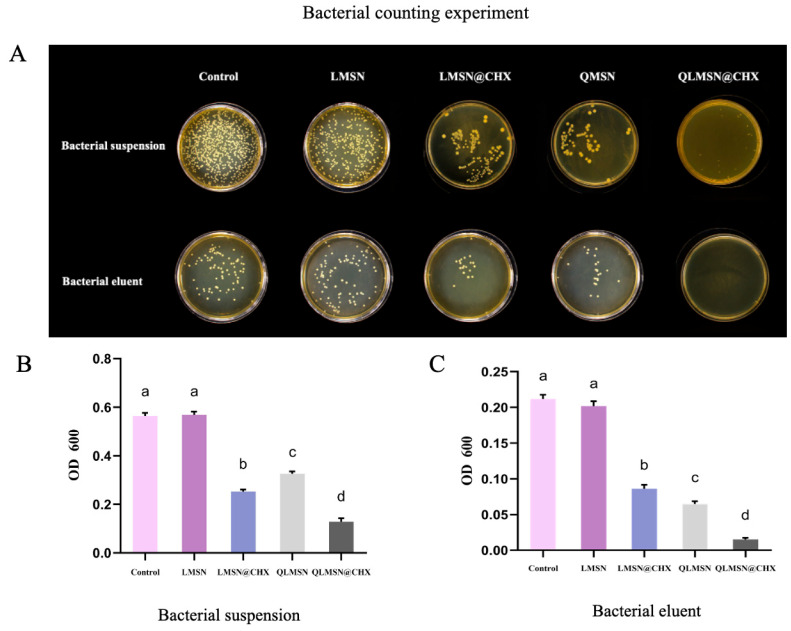
The antimicrobial performance of different resin components was validated using the plate colony counting method (**A**) and turbidity counting method (**B**,**C**). (Identical letters indicate no significant difference, whereas different letters represent statistically significant differences).

**Figure 5 ijms-26-06172-f005:**
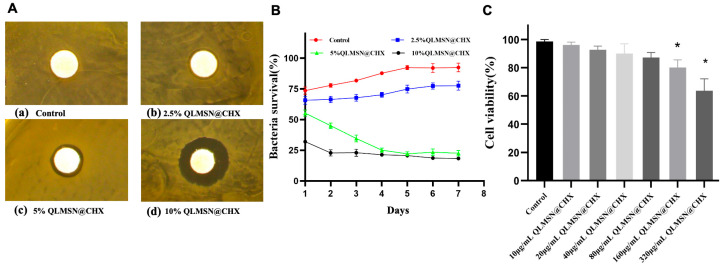
Antibacterial properties of resin containing QLMSN@CHX and cytotoxicity analysis of QLMSN@CHX. (**A**) Antibacterial properties of resin with different mass fractions of QLMSN@CHX. (**B**) MTT analysis of QLMSN@CHX at different concentrations. (**C**) Cytotoxicity evaluation of QLMSN@CHX at different concentrations using a human dental pulp mesenchymal stem cell line (CCK-8 assay) (* *p* < 0.05).

**Figure 6 ijms-26-06172-f006:**
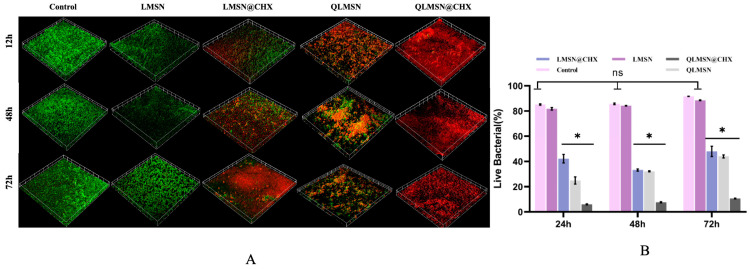
Antibacterial effects of different resins on Streptococcus mutans biofilms. (**A**) The 3D reconstruction of multispecies biofilms using confocal laser scanning microscopy, with live bacteria stained green and dead bacteria stained red. (**B**) Fluorescence quantification ratios of live and dead bacteria were calculated based on three random observation points of the bacterial biofilms. Data are expressed as mean ± standard deviation (* *p* < 0.05).

**Figure 7 ijms-26-06172-f007:**
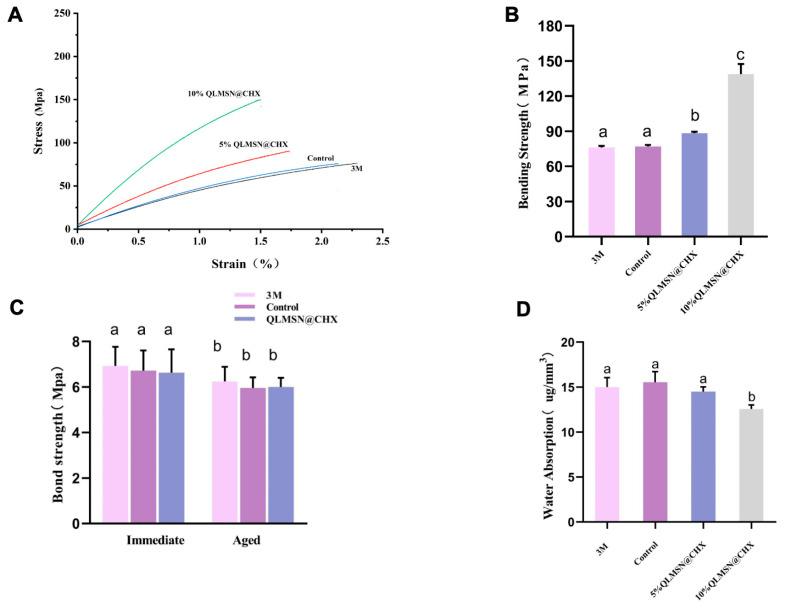
Mechanical and bonding properties of orthodontic adhesives: unmodified resin (Control), 3M commercial composite, 5 wt% QLMSN@CHX composite, and 10 wt% QLMSN@CHX composite. (**A**) Elastic modulus under three-point bending tests. (**B**) Bending strength. (**C**) Immediate and aged (30-day) shear bond strength to enamel. (**D**) Water absorption rates after 24 h immersion. (The same letter indicates no significant difference; different letters indicate a statistical difference).

**Table 1 ijms-26-06172-t001:** Resin residual index ARI frequency distribution (*n* = 30 per group).

			ARI			
	0	1	2	3	M (P25, P75)	*p*
3M	1	4	17	8	2 (2, 3)	a
Control	6	8	7	9	1 (2, 3)	a
5 wt% QLMSN@CHX	1	2	18	19	2 (2, 3)	a

Groups labeled with the same letter indicate no significant difference (*p* > 0.05).

## Data Availability

The data that support the findings of this study are available from the corresponding authors upon reasonable request.

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
