# Peer review of "Dual-Mode Antibacterial Orthodontic Composite: Contact-Killing QACs and Sustained CHX Release via Large-Pore Mesoporous Silica Nanoparticles"

_ijms, 2025, doi:10.3390/ijms26136172_

Round 1

Reviewer 1 Report

Comments and Suggestions for Authors

The present study focuses on the development of a dual-mode antibacterial adhesive by integrating 3-(trimethoxysilyl)-propyldimethylocta-decyl ammonium chloride (QACs), a quaternary ammonium salt, and chlorhexidine (CHX) in large-pore mesoporous silica nanoparticles (LMSN). The advanced antibacterial agent could act against bacteria by utilizing releasable (QACs) and non-releasable (CTAB) agents in combination.

In terms of the research structure, it is an in-depth approach, where the study starts with the synthesis of starting materials and ends with the development of the final dental formulations, while testing the properties of all products at all stages. The study extends to different research areas (synthesis and modification of mesoporous particles, drug release, properties of composite resins, etc.) resulting in remarks on both the different research areas and the structure of the work. More specifically:

  1. Diagrams in Figure 3 are very small and can’t be read. Authors should consider using larger diagrams. Moreover, diagrams 3B and 3C (Figure 3) are in place of each other. Their position needs to be corrected. Furthermore, probably the “Fig. 3E” phrase should be corrected to “Fig. 3D” in line 147.
  2. Figures 4-7 show the same image as Figure 1. Correct images should be inserted. Reviewing of corresponding paragraphs was impossible without the right Figures.
  3. Authors should cross-check the bibliographic sources with the references in the text. For example, citation No. 34 doesn’t correspond to the work of Sideridou et al. (lines 317-319), nor citations No. 38, 40, and 52 correspond to the works of Rodriguez, Karabela, and Zhang (line 506), respectively.
  4. According to authors, the synthesis of LMSN (lines 450-451) was conducted according to previous methods (ref. No. 51). However, the steps followed in this study are quite different from the reference, so more details about this methodology should be reported. For example, the mass of TEOS added, the reaction temperature during the first 8 hours of sealed reactor (was the reactor an autoclave?), the mass of TMB, the time of hydrothermal treatment at 80 °C, and the organic phase (CTAB+TMB) removal method (if it was removed). All these parameters play a crucial role in the morphology and textural properties of mesoporous silicas.
  5. Figure 2A shows the TEM image of MSNs before the addition of TMB, as indicated. The collection method and the treatment of nanoparticles at this stage (before TEM) should be mentioned.
  6. TMB is typically added during the template formation stage, before the introduction silica precursor (TEOS), in order to interact with the surfactant micelles (CTAB) and enlarge them. In this study, TMB is added 8 h after the addition of TEOS, so it can be assumed that the silica walls have already been formed around the surfactant micelles and the following hydrothermal treatment contributes to the increase of wall thickness by orienting and connecting the hybrid silica/micelle structures, without affecting the pore size. So, does the addition of TMB result in pore size increase? TEM images of Figure 2 don’t help to understand. Authors should consider providing higher magnification images or, preferably, N2 physisorption measurements to investigate the changes in the porosity of the particles. Moreover, LMSN of Figure 2B seem to have altered surface, compared to MSN, and not larger pores. In addition, their particle size seems to be smaller than that of MSN (Figure 2A), indicating surface alteration than pore enlargement. On the other hand, maybe this is a case of secondary penetrating channels or hollow vesicle structure. It is important for all these observations to be addressed in the final work.
  7. Authors should consider providing Encapsulation Efficiency and Loading Capacity values for CHX and LMSN.

The rest parts of the present work were not reviewed properly due to lack of information.

Author Response

Reviewer #1:

Comment 1:

Diagrams in Figure 3 are very small and can’t be read. Authors should consider using larger diagrams. Moreover, diagrams 3B and 3C (Figure 3) are in place of each other. Their position needs to be corrected. Furthermore, probably the “Fig. 3E” phrase should be corrected to “Fig. 3D” in line 147.

Response:

We have carefully addressed the issues regarding the size and resolution of Figure 3 by completely redesigning Figure 3A. Additionally, in strict accordance with the reviewers' comments, we have made corrections to the positioning of Figure 3D and 3E(lines180-184)

Comment 2:

Figures 4-7 show the same image as Figure 1. Correct images should be inserted. Reviewing of corresponding paragraphs was impossible without the right Figures.

Response:

We sincerely apologize for the formatting errors in Figures 4-7 caused by network issues during the submission process. We have now carefully updated these figures according to the corresponding experimental results, and respectfully request the reviewers' kind re-examination.

Comment 3:

Authors should cross-check the bibliographic sources with the references in the text. For example, citation No. 34 doesn’t correspond to the work of Sideridou et al. (lines 317-319), nor citations No. 38, 40, and 52 correspond to the works of Rodriguez, Karabela, and Zhang (line 506), respectively.

Response:

We have meticulously corrected the citation errors as follows: Replaced with the accurate reference to Sideridou et al. (line 358). Updated references for Rodriguez, Karabela, and Wilson (Lines 576-577), rectifying the attribution from "Zhang" to the first author Wilson.

Comment 4:

According to authors, the synthesis of LMSN (lines 450-451) was conducted according to previous methods (ref. No. 51). However, the steps followed in this study are quite different from the reference, so more details about this methodology should be reported. For example, the mass of TEOS added, the reaction temperature during the first 8 hours of sealed reactor (was the reactor an autoclave?), the mass of TMB, the time of hydrothermal treatment at 80 °C, and the organic phase (CTAB+TMB) removal method (if it was removed). All these parameters play a crucial role in the morphology and textural properties of mesoporous silicas.

Response:

We have corrected the citation of references for the LMSN preparation process in the text[1] (line 513). Detailed specifications including reaction temperature, precise mass of TMB, and hydrothermal treatment duration have been supplemented in the methodology section (lines 515-519). The synthesis was conducted in a high-pressure autoclave reactor.

Comment 5:

Figure 2A shows the TEM image of MSNs before the addition of TMB, as indicated. The collection method and the treatment of nanoparticles at this stage (before TEM) should be mentioned.

Response:

We have supplemented the TEM results in Figure 2, with Figure 2C specifically showing ultrathin cross-sectional images of QAS-functionalized LMSN (QLMSN). The sample pretreatment protocols and TEM preparation procedures have been added to the characterization methods section of the manuscript. (lines538-541)

Comment 6:

TMB is typically added during the template formation stage, before the introduction silica precursor (TEOS), in order to interact with the surfactant micelles (CTAB) and enlarge them. In this study, TMB is added 8 h after the addition of TEOS, so it can be assumed that the silica walls have already been formed around the surfactant micelles and the following hydrothermal treatment contributes to the increase of wall thickness by orienting and connecting the hybrid silica/micelle structures, without affecting the pore size. So, does the addition of TMB result in pore size increase? TEM images of Figure 2 don’t help to understand. Authors should consider providing higher magnification images or, preferably, N2 physisorption measurements to investigate the changes in the porosity of the particles. Moreover, LMSN of Figure 2B seem to have altered surface, compared to MSN, and not larger pores. In addition, their particle size seems to be smaller than that of MSN (Figure 2A), indicating surface alteration than pore enlargement. On the other hand, maybe this is a case of secondary penetrating channels or hollow vesicle structure. It is important for all these observations to be addressed in the final work.

Response:

TMB, as a commonly utilized pore-expanding agent, functions through the following mechanism: after mesoporous silica synthesis but prior to surfactant removal, hydrophobic TMB molecules are forced into the pore channels of micellar structures under hydrothermal conditions (high temperature/pressure). These molecules interact with the hydrophobic tails of surfactants (e.g., CTAB), thereby expanding the pore diameter. Subsequent removal of both the expanding agent and surfactants yields large-pore nanoparticle structures[2].[PMID: 21452883].

The preparation method and synthesis conditions for LMSN in this study align precisely with our previously validated protocols (referenced in line 513 of the original manuscript). Our prior work has conclusively demonstrated that LMSN samples prepared via this method exhibit:

Typical Type IV isotherms with hysteresis loops in N₂ physisorption analysis, attributable to "slit-shaped" pores formed between internal lamellae rather than surface deformation of LMSN [1]. [PMID: 26657191]

Structural consistency across multiple batches, as confirmed by TEM characterization.

In response to the reviewer's suggestion, we performed resin embedding and ultrathin sectioning of QLMSN samples to visualize internal pore architecture. The TEM results (Fig. 1C) corroborate the structural features observed in our previous studies. However, due to methodological prioritization in experimental design, we did not repeat N₂ physisorption analysis for pore size distribution quantification.

Regarding the secondary penetration channels or hollow vesicular structures noted by the reviewer, we hypothesize these may correlate with localized TMB concentration gradients during hydrothermal treatment. While this phenomenon warrants further investigation, detailed mechanistic analysis exceeds the scope of the current manuscript.

Comment 7:

Authors should consider providing Encapsulation Efficiency and Loading Capacity values for CHX and LMSN.

Response:

Dear Reviewer, thank you for your valuable suggestions regarding the evaluation of drug-loading performance. We appreciate the opportunity to clarify the following points about the encapsulation efficiency (EE%) and loading capacity (LC%) of chlorhexidine (CHX):

  1. Limitations of the current study:

This research prioritized the design and functional validation of the dual antibacterial mechanisms (contact-killing + release-killing). While we recognize the critical importance of EE% and LC% for comprehensive drug delivery characterization, systematic quantification of these parameters was not incorporated into the initial experimental design. We acknowledge this as a limitation of the current manuscript.

  1. Supporting indirect evidence

Although quantitative EE% and LC% data are unavailable, multiple lines of evidence collectively confirm the successful CHX loading and functional efficacy: (1) characteristic peak shifts in FTIR spectra (Figure 3A), (2) The total mass loss of QLMSN@CHX was 32.4%, which is 7.6% higher than that of QLMSN in TGA analysis (Figure 3B) (lines 170-172), (3) significantly enhanced antibacterial effects (expanded inhibition zones in Figure 5A), and (4) confocal microscopy results of antibacterial activity (Figure  6A). These consistent findings from different analytical approaches strongly support our conclusions.

We will prioritize establishing standardized quantitative drug loading assessment methods in our follow-up studies to address this methodological aspect more comprehensively.

Reviewer 2 Report

Comments and Suggestions for Authors

The current manuscript focuses on a very interesting topic. Othodontic appliances and materials can have significant impact on dental structures long term health. The study is well designed and conducted and the results are adequately presented. Probably due to by minor editing errors the images displayed in figures 4, 5, 6 are repetitive (and the same with figure 1) and not related to the description . Author should modify this aspect. 

Author Response

Reviewer #2:

Comment 1:

The current manuscript focuses on a very interesting topic. Orthodontic appliances and materials can have significant impact on dental structures long term health. The study is well designed and conducted and the results are adequately presented. Probably due to by minor editing errors the images displayed in figures 4, 5, 6 are repetitive (and the same with figure 1) and not related to the description. Author should modify this aspect.  

Response:

We appreciate the reviewer's positive evaluation of our work. We sincerely apologize for the formatting errors in Figures 1-7 that occurred due to network connectivity issues during the submission process. These figures have now been carefully revised and the corrected manuscript has been resubmitted to the system.

Reviewer 3 Report

Comments and Suggestions for Authors

The article presents a modern approach to modifying dental materials to give them antimicrobial properties. However, it contains several inaccuracies that should be addressed. The most significant issue is the absence of numerical data in the presentation of microbiological test results.

Specific comments:

The quaternary ammonium salt (QAC) used is specified only in the Materials section, near the end of the manuscript. This information should be presented earlier, preferably in the abstract or introduction.

The chemical structure of the QAC should be shown in a scheme. In addition, the chemical reactions for the surface modification of the silica nanoparticles with QAC and the anchoring reaction of the QAC into the orthodontic material matrix should be illustrated.

Line 38. It is unclear whether silver nanoparticles (AgNPs) and zinc oxide nanoparticles (ZnO) should be classified as leachable biocides. A separate citation is needed to support release behavior, distinct from chlorhexidine.

Line 98. The abbreviation "LMSN" appears for the first time without a prior definition.

Line 131. The -CH moiety does not represent a methyl group; a methyl group is denoted as -CH₃. This should be corrected.

Lines 142-150. The explanation provided in these lines lacks proper citations. Relevant references must be added.

Figure 3a. Signals corresponding to bond vibrations involving the quaternary nitrogen are not described in the FTIR spectrum. This requires completion to confirm the anchoring of QAC.

Figure 4 does not present experimental results but it is only a general scheme. Similarly, Figures 5 and 6 are repetitive and do not present actual experimental data. Experimental results for antibacterial and cytotoxicity tests should be shown in a clear format—preferably in tables or graphs—with precise descriptions, including measured values, representative images (where applicable), and statistical analyses.

The discussion on water absorption results could be strengthened by relating them to the thermogravimetric analysis results. In particular, the mass loss observed around 100 °C in TGA thermogram, which was attributed to water loss, should be discussed regarding the determined water absorption values.

Author Response

Reviewer #3:

Comment 1:

The quaternary ammonium salt (QAC) used is specified only in the Materials section, near the end of the manuscript. This information should be presented earlier, preferably in the abstract or introduction.

Response:

We sincerely appreciate the reviewer's constructive suggestion. In response, we have supplemented detailed descriptions regarding the QACs (Quaternary Ammonium Compounds) in the revised manuscript, which has helped better clarify the core research concept and experimental rationale in introduction. (lines 82-93)

Comment 2:

The chemical structure of the QAC should be shown in a scheme. In addition, the chemical reactions for the surface modification of the silica nanoparticles with QAC and the anchoring reaction of the QAC into the orthodontic material matrix should be illustrated.

Response:

We sincerely appreciate the reviewer's constructive suggestion. In the revised manuscript, we have added a schematic diagram illustrating the quaternization modification of silica nanoparticle surfaces by QAC, along with the chemical structure of QAC, which are now presented in Figure 1B.

The study follows a two-step approach: first, the antibacterial groups of QAC are anchored onto the nanoparticle surfaces to form QLMSN@CHX. Subsequently, these antimicrobial QLMSN@CHX particles are incorporated as fillers into the orthodontic resin matrix. Since the quaternization reaction between QAC and silica nanoparticles represents the core chemical transformation and QLMSN@CHX serves as the primary functional material, our flowchart primarily focuses on depicting this critical reaction.

To enhance clarity, we have supplemented additional explanatory notes for Figure 1 to better guide readers through the fundamental workflow of this study. The modifications have significantly improved the visual representation of our methodology while maintaining scientific accuracy. (line 94-104)

Comment 3:

Line 38. It is unclear whether silver nanoparticles (AgNPs) and zinc oxide nanoparticles (ZnO) should be classified as leachable biocides. A separate citation is needed to support release behavior, distinct from chlorhexidine.

Response:

We have supplemented supporting references in the relevant sections regarding silver nanoparticles (AgNPs), zinc oxide nanoparticles (ZnO), and chlorhexidine (CHX).(line37-39) The cited literature demonstrates that: (1) AgNPs exert antimicrobial effects through oxidative dissolution or ion exchange to release free Ag⁺, while ZnO generates Zn²⁺ and reactive oxygen species (ROS) via hydrolysis - both processes being significantly influenced by environmental chemical conditions (pH/ionic strength, etc.); (2) CHX typically releases through diffusion or ion exchange from carrier pores/matrices, with its release profile being more readily modulated by carrier characteristics. In our system, CHX is physically loaded into the abundant mesopores of LMSN, and subsequent release occurs upon contact with saliva during clinical application.

Comment 4:

Line 98. The abbreviation "LMSN" appears for the first time without a prior definition.

Response:

The full term "LMSN" is now highlighted in green upon its first appearance in the Introduction section. (line 94-95)

Comment 5:

Line 131. The -CH moiety does not represent a methyl group; a methyl group is denoted as -CH₃. This should be corrected.

Response:

Regarding the typographical error ("-CH"), we have corrected it to "-CH3" in the TGA characterization section. We sincerely appreciate the reviewer's meticulous attention to this detail. (line 165)

Comment 6:

Lines 142-150. The explanation provided in these lines lacks proper citations. Relevant references must be added.

Response:

This section presents the stability results of QMSN materials and the release profile of CHX, which serve as crucial indicators for validating the antimicrobial mechanisms. In the Discussion section, we have analyzed the relevant data concerning both QAS and CHX components. While we have provided literature citations[3](PMID: 39843087) (line 176)specifically for the methodological aspects of QAS stability testing, the remaining content presents objective experimental findings that do not require additional references.

Comment 7:

Figure 3a. Signals corresponding to bond vibrations involving the quaternary nitrogen are not described in the FTIR spectrum. This requires completion to confirm the anchoring of QAC.

Response:

We have carefully revised Figure 3A with a newly rendered FTIR spectrum that now includes clear labeling of characteristic peaks. The corresponding analytical results in the text have been comprehensively rewritten to provide robust evidence for the successful anchoring of QAS (Quaternary Ammonium Salts) to the nanoparticle surfaces. (lines 143-155)

Comment 8:

Figure 4 does not present experimental results but it is only a general scheme. Similarly, Figures 5 and 6 are repetitive and do not present actual experimental data. Experimental results for antibacterial and cytotoxicity tests should be shown in a clear format—preferably in tables or graphs—with precise descriptions, including measured values, representative images (where applicable), and statistical analyses.

Response:

We sincerely apologize for the significant formatting errors that occurred in the originally submitted manuscript due to network connectivity issues during the upload process. We have carefully reviewed and corrected the entire manuscript, with particular attention to Figures 4-7, for which we have supplemented additional supporting data to strengthen our findings.

Comment 9:

The discussion on water absorption results could be strengthened by relating them to the thermogravimetric analysis results. In particular, the mass loss observed around 100 °C in TGA thermogram, which was attributed to water loss, should be discussed regarding the determined water absorption values.

Response:

In response to the reviewer's valuable suggestion, we have significantly enhanced the Discussion section by incorporating a thorough analysis of the mechanistic correlation between TGA results and water absorption properties. This addition provides important insights into how material hydrophilicity/hydrophobicity influences functional performance. (lines 451-468)

Round 2

Reviewer 1 Report

Comments and Suggestions for Authors

I would like to thank the authors for the time and effort they took to make the changes according to the comments. There are no additional comments and the text is proposed for publication as is. Congratulations for your work

Author Response

Thank you for the judges' recognition.

Reviewer 3 Report

Comments and Suggestions for Authors

Thank you for the revised version. I accept all improvements.

Author Response

Thank you for the judges' recognition.